# Linguistic Analysis for Identifying Depression and Subsequent Suicidal Ideation on Weibo: Machine Learning Approaches

**DOI:** 10.3390/ijerph20032688

**Published:** 2023-02-02

**Authors:** Wei Pan, Xianbin Wang, Wenwei Zhou, Bowen Hang, Liwen Guo

**Affiliations:** 1Key Laboratory of Adolescent Cyberpsychology and Behavior (CCNU), Ministry of Education, Wuhan 430079, China; 2School of Psychology, Central China Normal University, Wuhan 430079, China; 3Key Laboratory of Human Development and Mental Health of Hubei Province, Wuhan 430079, China

**Keywords:** depression, suicidal ideation, linguistic analysis, regression, topic modeling, Weibo

## Abstract

Depression is one of the most common mental illnesses but remains underdiagnosed. Suicide, as a core symptom of depression, urgently needs to be monitored at an early stage, i.e., the suicidal ideation (SI) stage. Depression and subsequent suicidal ideation should be supervised on social media. In this research, we investigated depression and concomitant suicidal ideation by identifying individuals’ linguistic characteristics through machine learning approaches. On Weibo, we sampled 487,251 posts from 3196 users from the depression super topic community (DSTC) as the depression group and 357,939 posts from 5167 active users on Weibo as the control group. The results of the logistic regression model showed that the SCLIWC (simplified Chinese version of LIWC) features such as affection, positive emotion, negative emotion, sadness, health, and death significantly predicted depression (Nagelkerke’s *R*^2^ = 0.64). For model performance: *F-measure* = 0.78, area under the curve (*AUC*) = 0.82. The independent samples’ *t*-test showed that SI was significantly different between the depression (0.28 ± 0.5) and control groups (−0.29 ± 0.72) (*t* = 24.71, *p* < 0.001). The results of the linear regression model showed that the SCLIWC features, such as social, family, affection, positive emotion, negative emotion, sadness, health, work, achieve, and death, significantly predicted suicidal ideation. The adjusted *R*^2^ was 0.42. For model performance, the correlation between the actual SI and predicted SI on the test set was significant (*r* = 0.65, *p* < 0.001). The topic modeling results were in accordance with the machine learning results. This study systematically investigated depression and subsequent SI-related linguistic characteristics based on a large-scale Weibo dataset. The findings suggest that analyzing the linguistic characteristics on online depression communities serves as an efficient approach to identify depression and subsequent suicidal ideation, assisting further prevention and intervention.

## 1. Introduction

Depression is the commonest psychiatric disorder, with an estimated 3.8% of the population affected, including 5.0% of adults [1]. The research showed that the lifetime prevalence of depression reached 20% [2,3]. Depression is underdiagnosed [4]. For instance, the recognition accuracy of depression by general practitioners was unsatisfying [5], and both the sensitivity and specificity of the diagnostic performance of machine learning models were proven to be higher than that of health-care professionals [6]. Therefore, objective and accurate tools to identify cases of depression will have major clinical benefits.

Social media platforms, such as Twitter, Facebook, and Weibo, are popular places where individuals express and record their personalities, feelings, moods, thoughts, and behaviors [7]. Text mining on social media is useful in detecting cases of depression [8]. For example, based on Weibo posts from 180 users, Wang et al. [9] found that the quantity of emoticons and first-person pronouns significantly predicted depression. This study employed three types of classifiers and achieved around 80% accuracy. Cheng et al. [10] included 974 Weibo users and utilized computerized language analysis methods to assess emotional distress. In this research, achievement-related words and work-related words were significantly associated with depression, but the classifiers did not achieve a satisfying performance. Moreover, Ricard et al. [11] investigated the utility of user-generated content, such as posts, in predicting depression among 749 participants. The results showed that the model trained on user-generated online data did not achieve statistical significance (*AUC* = 0.63, *p* = 0.11). Overall, the predicting accuracies remained inconsistent among the previous studies. Liu et al. [7] reviewed studies that applied machine learning methods to text data on social media to detect depressive symptoms from January 1990 to December 2020. They pointed out that the previous studies were limited by small sample sizes and suggested that large-scale dataset could facilitate high accuracy in predictive applications. Hence, it is necessary to develop depression identification models on a large sample size on social media.

Online depression communities (ODCs) are powerful forums for self-disclosure and social support seeking around mental health issues. Many users who struggle with depression gather there. Tadesse et al. [12] detected factors that revealed the depression attitude on Reddit. The data contained 1293 depression-indicative posts from relatively large depression subreddits and 548 non-depressed posts, and the classifier for depression detection reached a 0.93 F1 score. A recent study [13] coded depression-related symptoms based on Weibo posts from a large sample of 19,634 ODC members to better understand the content of these symptoms and their associations. The network analysis indicated that suicidality was the most central symptom, and there was a strong correlation between low self-evaluation and self-blame. This study provided an in-depth understanding of depression. It appears that ODCs provide depression detection with consistently high predicting accuracy and explainable information.

Suicide is a core symptom of depression [7]. Suicide can be divided into three levels: SI, suicide attempts, and completed suicide. According to Beck et al. [14], SI is a desire or plan to commit suicide without a real attempt yet, and it is an important indicator for suicide risk assessment. Research has also identified SI as one of the strongest predictors of suicide attempts [15,16,17]. Hence, SI supervision is the key to effective suicide prevention among depressed patients.

The digital footprint for SI is also tractable online. People with SI often seek help or leave suicide notes on social media before attempting suicide [18]. Jashinsky et al. [19] found that there was a strong correlation between the proportion of suicide risk-related tweets and the actual suicide rates, validating social media as a potential dataset for studying suicide. Aldhyani et al. [20] built a suicidal ideation detection system using Reddit datasets, word-embedding approaches, such as TF-IDF and Word2Vec, for text representation, and hybrid deep learning and machine learning algorithms for classification. Models achieved 95% suicidal ideation detection accuracy. Similar studies were also conducted on Weibo. For instance, Gu et al. [21] extracted text features from Weibo data and built a suicide risk prediction model to predict four dimensions of the Suicide Possibility Scale—hopelessness, suicidal ideation, negative self-evaluation, and hostility—and all achieved adequate performance. Liu et al. [22] detected suicidal ideation on Weibo using an ensemble method; this approach was assessed with a dataset formed from 40,222 posts. By integrating the best classification models of single features and multi-dimensional features, the model achieved 79.20% F1-scores. These studies were aimed at investigating suicide in general. In fact, it was suggested that predicting suicidal behavior using social media analytics should be undertaken carefully because each person with suicidal behavior has different risk factors than others [20]. Depression is the most common psychiatric disorder in people who commit suicide. Depression had a higher suicide rate relative to the general population [23,24].

In studies that investigated suicide and mental health status, such as depression, according to a recent systematic literature review of 96 relevant studies concerning suicide and depression detection on social media [25], only several studies detected both the depression level and suicide or self-harm from social media content [26,27,28,29,30,31,32,33,34,35,36]. A study [10] also investigated suicide risk and its risk factors, such as depression and anxiety, on Weibo. To the best of our knowledge, few prior studies have forecasted the SI of depressed patients on social media. As failing to identify a person with high suicide risk could lead to loss of life, a more targeted strategy to precisely identify people with a high suicide risk is advantageous for suicide prevention.

To fill in these gaps, this research focused on identifying depression and the subsequent SI symptom with posts from an ODC and random posts from the control group on Weibo. Our purpose was to establish machine learning models based on linguistic features to identify depressed patients that are at a high risk of suicide, so that intervention could step in promptly. As other machine learning models, such as neural networks, are opaque for researchers or clinicians to interpret [37], we employed regression methods, i.e., the logistic regression model and linear regression model, in this research. Regression models offer multiple parameters to help clarify the underlying mechanisms, which are frequently used in text mining on social media [38]. Topic modeling was also exploited to assist the understanding of the Weibo content.

## 2. Materials and Methods

### 2.1. Participants and Data Collection

The data were collected from Weibo, a Chinese microblogging website and one of the biggest social media platforms. First, we located the largest ODC on Weibo called the depression super topic community (DSTC). As of November 2022, it had more than 305,000 subscribers, 901,000 postings, and 3.09 billion hits. An ODC post is similar to a tweet with the hashtag on Twitter. However, it is noteworthy that the DSTC is managed by website hosts to guarantee that all posts must be depression-related. For example, posts concerning online dating, sharing unprofessional online self-diagnosis tools, and all kinds of advertisements (i.e., depression medicine, psychological consulting institutions, and so on) will be removed. Simple emotion-outpouring (instead of depression-related) posts also violate posting restrictions and will not show in the DSTC. Most of the DSTC posts were accompanied by formal clinical diagnoses, and the rest was from potential depressed patients that had not been diagnosed but needed professional help.

The data were acquired utilizing “Houyi”, a professional web scraping software. First, 11,142 posts were collected from the DSTC (posts from the DSTC were set as the DSTC group). According to account information of these posts, we then located their Weibo homepage and acquired 487,251 posts from 3196 users, set as the depression group in the present study. We then downloaded 357,939 random posts from 5167 Weibo users outside of the DSTC and other mental illness-related online communities, which formed the control group in this study.

The obtained data included users’ (1) profile information, (2) online behaviors, and (3) Weibo posts. User privacy was protected in this research.

### 2.2. Psychological Lexicons

The Simplified Chinese Version of LIWC (SCLIWC). SCLIWC was developed by the Computational Cyber Psychology Laboratory at the Institute of Psychology, Chinese Academy of Sciences [39]. The present study used the SCLIWC software to extract psychologically meaningful word features and their frequencies in Weibo posts. The effectiveness of this method has been verified in previous studies [40,41,42,43]. A total of 102 SCLIWC features were extracted. The proportion of frequency of each psychological word category was calculated for each Weibo user [44,45]. Then, standardization was applied.

Chinese Suicide Dictionary (CSD). We used the CSD [46] to calculate the SI scores of Weibo users. The CSD includes 586 keywords related to SI, such as “牵挂” (worry), “轮回” (reincarnation), and “永别” (part forever). Previous studies have proven its effectiveness in detecting SI [46,47,48]. In the SI calculation, the frequency of dictionary words in each post was counted, and the weights of those dictionary words in each post were summed up [46]. If the total score of one post was up to three, then this post was recognized as a post with SI. For each user, the proportion of Weibo posts with SI was considered as his/her level of SI.

### 2.3. Data Analysis

#### 2.3.1. Logistic Regression Modeling

We performed logistic regression analysis on the extracted SCLIWC features to investigate whether these features could help distinguish the depression group from the control group. There were 8363 users, 3196 of which were depression-related. We randomly split the data into training set and test set with a ratio of 7:3. Logistic regression model was built on the training set with the glm function in R [49,50,51,52]. We then examined the model performance on the test set. To avoid multicollinearity, we checked variance inflation factor (VIF) for each SCLIWC word feature. A VIF value > 10 was considered as indicating multicollinearity [53], and the corresponding SCLIWC features were removed from the model.

#### 2.3.2. Linear Regression Modeling

We built linear regression models using SCLIWC features to predict SI. We chose SCLIWC features that significantly classified the depression group and the control group as independent variables. Training set and test set were the same as datasets in the logistic regression model. We performed linear regression model analysis on the training set with the lm function in R [54,55] and applied the model on the test set. For model performance, we calculated the correlation between the actual SI values and the predicted SI values of the test set. To avoid multicollinearity, we also performed VIF examination on the SCLIWC features.

#### 2.3.3. Topic Modeling

To describe the content of posts for each group (the depression group, the control group, and the DSTC group) for a better understanding, we further performed topic modeling to extract their topics, which yielded an abstract of the topics for each group. We used the Jieba tool, a Chinese word segmentation package on Python, to cut users’ original microblog content into individual words. We used latent Dirichlet allocation (LDA), a Bayesian inference method, to discover topics from given corpora [56].

## 3. Results

### 3.1. Logistic Regression Modeling

The SCLIWC features of Article, enPast, enParent, enFuture, and NumAtMention were removed as these five columns were all NAs, and the SCLIWC features of Funct, Pronoun, PPron, TenseM, and CogMech were removed as their VIF values were over 10.

The final results contained 92 SCLIWC features. Table 1 displays features that were significant in the logistic regression models. The results showed that the SCLIWC features of affection, positive emotion, negative emotion, sadness, health, and death significantly predicted depression (Nagelkerke’s R^2^ = 0.64). For model performance, sensitivity = 0.74, specificity = 0.9, accuracy = 0.84, precision = 0.82, recall = 0.74, and F-measure = 0.78. The goodness-of-fit test showed that the AUC value was 0.82. For full model information, please check the Appendix A.

### 3.2. Linear Regression Modeling

The independent sample *t*-test showed that the SI scores between the depression (0.28 ± 0.5) and control groups (−0.29 ± 0.72) were significantly different (*t* = 24.71, *p* < 0.001). The results from the linear regression model showed that the SCLIWC features, including social, family, affection, positive emotion, negative emotion, sadness, health, work, achieve, and death, significantly predicted SI (see Table 2). The VIF test indicated no multicollinearity issues. The adjusted *R^2^* was 0.42. For model performance, the correlation between the actual and predicted SI on the test set was significant (*r* = 0.65, *p* < 0.001).

### 3.3. Topic Modeling

We extracted five topics for each group, each containing 10 key words, and the max. iteration number was set to 10. To better understand these extracted topics for each group, all authors reviewed these topics and made agreements on one abstract for each group (see Table 3, Table 4 and Table 5).

## 4. Discussion

This research identified depression and the subsequent suicidal ideation symptom using the linguistic characteristics of posts on Weibo. The results exhibited the sufficiency of machine learning models in detecting depression and subsequent suicidal ideation.

To start with, we employed the logistic regression model to examine whether the linguistic characteristics significantly separated the depression group and the control group and to identify depression-related SCLIWC features. The results showed that features concerning I, we, social, family, positive and negative emotions, sadness, health, work, achieve, money, and death significantly differentiated the depression group and the control group, and the contribution of these features was 64%. The model performance is promising (*F-measure* = 0.78, *AUC* = 0.82). Such results are consistent with previous research. People with depression usually use more first-person singular pronouns and negative emotion words in their postings [57]. The frequent use of first-person singular pronouns is a marker of self-awareness [58,59]. High self-awareness is a known psychological attribute in depression [60]. Emotions, especially negative emotions, were proven to be a key symptom of depression. Specifically, sadness was found to be the main emotion expressed in postings of online depression communities [13].

Next, we established the linear regression model to examine if the expression of depression-related linguistic characteristics significantly predicted SI. The results showed that the SCLIWC features, including social, family, positive/negative emotions, sadness, health, work, achieve, money and death, significantly predicted SI. The contribution of these features to SI was 42%. The model performance was satisfactory as the correlation between actual SI and predicted SI reached 0.65 (*p* < 0.001). SI was significantly different between the depression and control groups. These results are in accordance with previous research. A study [61] used LIWC to conduct a comparative analysis of suicidal and non-suicidal Reddit content. They found that users with suicidal ideation scored significantly higher in the negative emotions, such as anxiety and sadness, compared with average users. Similar results were found in other studies [62,63,64,65,66,67]. Research also suggested that the impact of negative emotions on SI was more direct than other factors, such as personality traits [68], adverse life events [66], and attitudes towards suicide [67].

It is worth noting that the significance of linguistic features in the logistics regression model and the linear regression model were highly consistent.

Firstly, social and family features were significant in both models. These results are congruent with the topic modeling results for the DSTC group, where family and friends were frequently mentioned, suggesting that depressed patients may experience social support issues, and according to the integrated motivational–volitional model of suicidal behavior theory [69], lack of social support can be a hazard factor for suicide.

Secondly, health-related linguistic features were also significant in both models. The online depression community is a place where patients share their mental health issues and treatment and seek help. Research has detected large numbers of health-related descriptions, such as sleeping issues, among depressed patients and people with SI [13,70]. Third, the work, achieve, and money-related features were significant in a negative direction in both models. Work and achievements are highly associated with higher self-esteem or self-efficacy [71,72]. According to Yao et al., depressed patients usually experience low self-evaluation and negative expectation [13]. Work and achievement may protect individuals from depression and further SI.

Finally, the tentative and discrepancy-related features were also significant in both models, while features, such as insight or cause-related words, were not significant in either model. These results may suggest problematic cognitive processing in the depressed patients. A review on the cognition abnormalities of major depressive disorder (MDD) found that indecisiveness was among the most troubling symptoms of MDD patients [73]. Moreover, there are two basic types of cognitive dysfunction observed in depression: (1) cognitive biases, which include distorted information processing or attentional allocation toward negative stimuli and away from positive stimuli, and (2) cognitive deficits, which include impairments in attention, short-term memory, and executive functioning [74]. Together with the negative emotion relevant results, we infer that depressed patients may go through high emotion arousal and irrational cognition. These evidences indicate that depressed patients may benefit from psychological therapies, such as relaxation training and cognitive interventions, to reduce their depression and SI symptoms.

Topic modeling results are consistent with the logistic and linear regression model results. The depression group, compared with the control group, had fewer entertainment-related expressions. Moreover, all posts in the DSTC group were highly depression-related. These results suggest that future depression and subsequent suicide intervention should pay more attention to depression-related online communities to precisely locate targeted populations that urgently need professional help.

This research has several limitations. First, we failed to report descriptive statistics of participants’ demographic information due to high missingness. Therefore, generalization of the results needs to be careful. Second, although we tried explaining the linguistic features that characterized depressed patients and their SI, most of the explanations were made by referring to previous research. Further work should take this research as a preliminary study and examine whether characteristics such as social support, emotions, self-evaluation, and cognition can be extracted from the LIWC features. As machine learning models are not intuitive for psychiatrists, future research should explore explainable machine learning models that link linguistic features to depression-related psychological factors. Moreover, replication work should test the classifiers on other online communities to check the model generalization ability to facilitate clinical applications of depression and the subsequent SI detection.

## 5. Conclusions

This research systematically investigated depression and subsequent SI-related linguistic characteristics based on a large-scale Weibo dataset. The machine learning models served as promising ways to efficiently identify depressed patients and their suicide risks. Future research should focus on building explainable machine learning models on ODCs and testing model generalization to facilitate the clinical detection and intervention of depression and the resulting suicide.

## Figures and Tables

**Table 1 ijerph-20-02688-t001:** Logistic regression model using the simplified Chinese version of LIWC (SCLIWC) features to classify the depression and control groups.

SCLIWC Features	*β*	*S.E.*	*Z*	*p*	*sig*
Intercept	−0.47	0.06	−8.00	0.00	***
I	1.02	0.10	10.52	0.00	***
We	0.14	0.06	2.14	0.03	*
Quantifier	−0.20	0.06	−3.59	0.00	***
Prepend	0.21	0.06	3.48	0.00	***
Specart	−0.30	0.06	−5.09	0.00	***
Multi-Functional	−0.26	0.09	−2.96	0.00	**
Social	0.23	0.08	2.80	0.01	**
Family	−0.16	0.06	−2.70	0.01	**
Affection	0.40	0.13	3.00	0.00	**
Positive Emotions	−0.25	0.10	−2.43	0.02	*
Negative Emotions	0.40	0.12	3.23	0.00	**
Sadness	0.41	0.09	4.44	0.00	***
Discrepancies	0.51	0.13	4.04	0.00	***
Tentative	0.34	0.10	3.41	0.00	***
Exclusive	−0.35	0.10	−3.57	0.00	***
Perceptual Processes	−0.40	0.11	−3.71	0.00	***
See	0.38	0.11	3.48	0.00	***
Hear	0.17	0.07	2.34	0.02	*
Health	0.70	0.09	7.95	0.00	***
Ingest	−0.26	0.09	−2.88	0.00	**
Relativity	−0.40	0.11	−3.80	0.00	***
Motion	0.32	0.07	4.89	0.00	***
Work	−0.19	0.06	−2.92	0.00	**
Achieve	−0.30	0.07	−4.33	0.00	***
Money	−0.21	0.08	−2.67	0.01	**
Death	0.36	0.08	4.37	0.00	***
Nonfluencies	0.21	0.08	2.60	0.01	**
Filler Words	−0.28	0.08	−3.57	0.00	***
Period	−0.15	0.07	−2.15	0.03	*
Comma	−0.28	0.06	−4.68	0.00	***
Semicolon	−0.29	0.08	−3.51	0.00	***
Exclamation	−0.22	0.07	−3.29	0.00	**
Dash	0.31	0.08	3.77	0.00	***
Quote	−0.27	0.08	−3.25	0.00	**
Parentheses	0.18	0.07	2.69	0.01	**
Other Punctuation	−0.17	0.07	−2.38	0.02	*
Word Count	1.00	0.07	14.54	0.00	***
Words Per Sentence	−0.59	0.20	−3.02	0.00	**
Rate of Dictionary Cover	−0.30	0.11	−2.82	0.00	**
Rate of Numerals	−0.70	0.09	−8.11	0.00	***
Rate Four Char Words	−0.19	0.08	−2.31	0.02	*
Rate of Latin Words	0.38	0.08	4.52	0.00	***
Number of Emotions	−0.76	0.10	−7.61	0.00	***
Number of Hashtags	0.42	0.08	5.42	0.00	***
Number of URLs	0.29	0.07	4.24	0.00	***

*** *p* < 0.001, ** *p* < 0.01, * *p* < 0.05.

**Table 2 ijerph-20-02688-t002:** Linear regression model using the simplified Chinese version of LIWC (SCLIWC) features to predict suicidal ideation (SI).

SCLIWC Features	*β*	*S.E.*	*T*	*p*	*Sig*
We	0.10	0.01	10.13	0.00	***
Quantifier	−0.04	0.01	−4.19	0.00	***
Prepend	0.03	0.01	2.65	0.01	**
Specart	0.03	0.01	3.22	0.00	**
Social	0.07	0.01	5.90	0.00	***
Family	0.02	0.01	1.98	0.05	*
Affection	0.07	0.02	3.26	0.00	**
Positive Emotion	−0.09	0.02	−5.54	0.00	***
Negative Emotion	0.15	0.02	8.39	0.00	***
Sadness	0.04	0.02	2.72	0.01	**
Discrepancies	0.05	0.02	3.46	0.00	***
Tentative	0.08	0.02	5.15	0.00	***
Exclusive	−0.09	0.02	−6.00	0.00	***
See	−0.10	0.02	−5.73	0.00	***
Hear	0.06	0.01	5.09	0.00	***
Health	0.11	0.01	8.59	0.00	***
Ingest	−0.09	0.01	−7.27	0.00	***
Relativity	−0.11	0.01	−10.52	0.00	***
Work	0.08	0.01	7.87	0.00	***
Achieve	−0.02	0.01	−2.13	0.03	*
Money	−0.07	0.01	−4.88	0.00	***
Death	0.12	0.01	9.01	0.00	***
Nonfluencies	0.05	0.01	3.99	0.00	***
Period	0.07	0.01	6.13	0.00	***
Comma	0.08	0.01	8.43	0.00	***
Exclamation	−0.02	0.01	−1.97	0.05	*
Parentheses	−0.03	0.01	−2.63	0.01	**
Other Punctuation	−0.11	0.01	−8.55	0.00	***
Word Count	0.12	0.01	10.56	0.00	***
Words Per Sentence	0.14	0.02	5.57	0.00	***
Rate of Dictionary Cover	0.09	0.02	5.37	0.00	***
Rate Four Char Words	0.12	0.01	9.76	0.00	***
Rate of Latin Words	−0.08	0.01	−5.44	0.00	***
Number of Emotions	0.06	0.02	3.66	0.00	***
Number of Hashtags	0.07	0.02	4.77	0.00	***
Number of URLs	−0.03	0.01	−4.24	0.00	***

*** *p* < 0.001, ** *p* < 0.01, * *p* < 0.05.

**Table 3 ijerph-20-02688-t003:** Topics and abstract of topics for the control group.

Abstract	Topics
(1) Common topics that are usually discussed on Weibo such as idolization and hot events such as the Olympic Games and New Year; (2) Common online behavior such as microblog forwarding.	Yuan Wang *, TF boys, microblog, Junkai Wang, video, broadcast, music, Qianxi, juvenile
	Full text, video, microblog, broadcast, China, hahahaha, America, hahaha, link, webpage
	Endeavour, Yixing Zhang, microblog, hip-hop, Zhan Xiao, video, broadcast, juvenile, studio, September 18th incident
	Microblog, forwarding, video, broadcast, full text, China, really, 10000 times, lottery, like
	Microblog, red envelope, cash, 2021, forwarding, New Year’s Eve, links, web pages, Fortune, voting

* Names occurring in topics belong to celebrities, similarly hereinafter.

**Table 4 ijerph-20-02688-t004:** Topics and abstract of topics for the depression group.

Abstract	Topics
(1) Common topics that are usually discussed on Weibo, such as music and idolization; (2) Mental health-related words such as depression and hope.	Xukun Cai, Jie Zhang, cover, year, 2019, music, 2020, new song, Minghao Huang, Xun Wei
	Microblog, forwarding, video, full text, playback, link, webpage, hahahaha, juvenile, lottery
	Depression, microblog, real, full text, forwarding, video, love, hope, life, feeling
	TF, boys, Yuan Wang, Qian Xi, Junkai Wang, band, Xia Ye, film, new album, fan club
	Dilraba, dear, Lu Bai, Glory, Jingjing, Badaling, Peacock, Deyun Society, Kowloon

**Table 5 ijerph-20-02688-t005:** Topics and abstract of topics for the depression super topic community (DSTC) group.

Abstract	Topics
(1) Negative emotions; (2) Depression treatment-related discussion; (3) Family and friends.	No, really, taking medicine, depression, doctor, depression, living, whether there is, fear, making me
	Really, feeling, happiness, life, emotion, disappointment, suffering, sadness, as if
	A little, hope, collapse, mom, sleep, someone, sometimes, good night, Er Ha (silly like a Siberian husky), friends
	Evening, leaving, happy, want to, understanding, a few days, friends, http, cn, cute
	Like, work, bad, world, hospital, more and more, if, tell, tomorrow, doctor

## Data Availability

Datasets of this study are available from the corresponding author on reasonable request.

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
