# Peer review of "Linguistic Analysis for Identifying Depression and Subsequent Suicidal Ideation on Weibo: Machine Learning Approaches"

_ijerph, 2023, doi:10.3390/ijerph20032688_

Round 1

Reviewer 1 Report

Title: “Linguistic Analysis for Identifying Depression and Its Accom- 2 panying Suicidal Ideation on Weibo: Machine Learning Ap- 3 proaches” I have read this Paper thoroughly and have some observations:

1.      Conclusion presentation is very poor. Please add future research direction in conclusion section with elaboration.

2.      Please highlight main contribution of the research in Abstract section.   

3.      Please add discussion section and sensitivity analysis with respect to major parameter.

4.      Comparative analysis presentation in the revised paper.

5.      Paper has some grammatical errors please re-check all grammatical mistake in the revised version.

6.      Introduction section has much non related citation. Please add some related citation.

Reviewer 2 Report

Some percentage (39%) of plagiarism has been detected on Turnitin.

The article needs some improvements for English. The dataset used is just one and related to a sub-community. 

It would be interesting to evaluate the study on several datasets from different social networks. 

Title promises machine learning approaches but actually only two are provided.

Conclusions reported confirm that this work simply validates previous results from other researches. So the novelty lies only in the fact that that previous study conclusions apply to this new Weibo dataset too.

So, further investigations on other datasets and/or models are needed.

Round 2

Reviewer 2 Report

Unfortunally most of the previous comments of my previous review were not take into consideration. 

The most relevant is related to the novelty of the work that, as already stated: "Conclusions reported confirm that this work simply validates previous results from other researches. So the novelty lies only in the fact that that previous study conclusions apply to this new Weibo dataset too."
